# Learning Curve for Ultrasound Assessment of Myometrial Infiltration in Endometrial Cancer Visualizing Videoclips: Potential Implications for Training

**DOI:** 10.3390/diagnostics13030425

**Published:** 2023-01-24

**Authors:** Enrique Chacón, Julia Marucco, Irene Martinez, Alba Monroy, Maria Victoria Laza, Aida Tomaizeh, Maria Ángela Pascual, Stefano Guerriero, Juan Luis Alcázar

**Affiliations:** 1Department of Obstetrics and Gynecology, Clínica Universitaria De Navarra, 31008 Pamplona, Spain; 2Centro de Diagnóstico TCBA, Buenos Aires C1177, Argentina; 3Department of Obstetrics, Gynecology and Reproductive Medicine, Hospital Universitario Virgen de Valme, 41014 Sevilla, Spain; 4Department of Obstetrics and Gynecology, Hospital Materno-Infantil, 06010 Badajoz, Spain; 5Department of Obstetrics, Gynecology, and Reproduction, Hospital Universitari Dexeus, 08028 Barcelona, Spain; 6Centro Integrato di Procreazione Medicalmente Assistita (PMA) e Diagnostica Ostetrico-Ginecologica, Azienda Ospedaliero Universitaria—Policlinico Duilio Casula, Monserrato, University of Cagliari, 09042 Cagliari, Italy

**Keywords:** training, ultrasound, myometrial infiltration, endometrial cancer

## Abstract

Background: Diagnostic accuracy for estimating myometrial infiltration by ultrasound in endometrial cancer requires experience. The objective of this study is to determine the learning curve (LC) for assessing myometrial infiltration in cases of endometrial cancer using transvaginal ultrasound (TVS). Methods: Five trainees (one staff radiologist and four fourth-year OB/GYN residents) participated in this study. All trainees had experience in performing TVS, but none of them had specific training on the assessment of myometrial infiltration. Trainees were given one specific lecture about the topic, and then they observed videoclips from 10 cases explained by the trainer. After this, all trainees visualized 45 videoclips of uterine ultrasound scans of endometrial cancer cases. The assessment of myometrial infiltration was based on the subjective impression. Definitive histology was used as a reference standard. Trainees stated whether myometrial infiltration was ≥50% or <50%. LC-CUSUM and standard CUSUM graphics were plotted to determine how many cases were needed to reach competence, allowing a mistake rate of 15%. Results: All trainees completed the study. LC-CUSUM graphics showed that three trainees reached competence at the 33rd, 35th and 36th case, respectively. All three of them kept the process under control after reaching competence. One trainee reached competence but did not maintain it in the cumulative analysis. One trainee did not reach competence. Conclusion: Our study suggests that 30–40 cases would be needed to be trained for assessing myometrial infiltration by TVS by visual interpretation of videoclips by most trainees.

## 1. Introduction

Endometrial cancer is the sixth most frequent form of cancer in women worldwide. In fact, it is the most common cause of gynecologic malignancy in developed countries, with an incidence of 12.9 per 100,000 women per year and a mortality rate of 2.4 per 100,000 women per year [1]. The most important prognosis features for endometrial cancer are tumor stage (according to International Federation of Gynaecology and Obstetrics classification), myometrial invasion, histological type and tumor grade. Among these, tumor invasion of 50% or more of the myometrial wall is associated with a higher risk of lymph node metastasis [2,3]. In fact, in well-differentiated endometroid type cancers, the probability of lymph node involvement increases from 4% to 15% if the tumor invades more than 50% of the myometrial wall [2]. The risk of relapse in these patients is higher, and they may require complete surgical staging with systematic pelvic and para-aortic lymph node dissection to define whether adjuvant therapy is advisable. Additionally, avoiding overtreatment is of paramount importance in these patients, especially taking into account the fact that a significant proportion of these women are elderly and may have comorbidities. On the contrary, women with low-grade endometrioid endometrial cancer and superficial myometrial infiltrations pose a very low risk of lymph node involvement, and systematic lymphadenectomy is not advised, as there is no beneficial therapeutic effect [4].

Current guidelines recommend the assessment of myometrial infiltration by magnetic resonance imaging or transvaginal ultrasound in the pre-operative work-up of women with endometrial cancer [4]. Both techniques offer similar diagnostic performance [5]. However, expertise is clearly needed when ultrasound is used [4]. Myometrial infiltration assessment by transvaginal ultrasound can be conducted using several approaches, namely examiner’s subjective impression, Karlsson’s method and Gordon’s method [6]. When using the examiner’s subjective impression, the depth of myometrial infiltration is estimated by assessing the point at which the myometrium–endometrium interface is not identified clearly and then by assessing the supposedly tumor-free myometrial wall at this point. The opposite myometrial wall is used for comparison, and, if marked asymmetry is found, deep (≥50%) infiltration is assigned; if myometrial thickness was similar in both myometrial walls, superficial (<50%) infiltration is assigned. In the Gordon’s approach, the ratio of the distance between the maximum tumor depth and total myometrial thickness is estimated. If the ratio is <50%, myometrial invasion is estimated as less than half of the myometrium; in contrast, if the ratio is >50%, it is classified as more than half of the myometrium (Figure 1).

Finally, in the Karlsson’s approach, the ratio between the maximum anteroposterior diameter of the endometrial lesion and the uterine anteroposterior diameter, both measured in the sagittal plane, is estimated. Similar to the Gordon’s approach, if the ratio is <50%, myometrial invasion is estimated as less than half of the myometrium; in contrast, if the ratio is >50%, it is classified as more than half of the myometrium (Figure 2).

Overall, all three methods have similar diagnostic performance [6]. However, in G1/G2 endometrioid carcinomas, the examiner’s subjective impression seems to be better than Karlsson’s and Gordon’s methods [7]. This makes it clear that examiner training is advisable for performing ultrasound examinations for this purpose. In fact, some studies have shown that training affects diagnostic performance of the examiner’s subjective impression when assessing myometrial infiltration in endometrial cancer [8].

Ultrasound training requires both “hands” and “eyes” training. In other words, it requires training in how to perform the ultrasound examination and training in how to interpret the images observed [9]. Our group has demonstrated the effectiveness of a structured training program for ultrasound examination in some areas of gynecological ultrasound such as adnexal masses [10], endometriosis [11] and uterine congenital anomalies [12].

Traditionally, performing a determined number of ultrasound scans has been the criterion to establish competence in gynecological ultrasound [13,14]. However, this method lacks objective evidence regarding individual competence. Statistical methods have been applied to analyze the interventional and diagnostic learning curve to overcome this issue. The cumulative summation (CUSUM) test and the LC-CUSUM test are currently good approaches to assess when competence is achieved [15].

There is a paucity of studies about the learning curve of sonographers performing transvaginal ultrasound assessments of uterine wall infiltration in women with endometrial cancer. To the best of our knowledge, there is only a recent study assessing the learning curve for evaluating myometrial infiltration by transvaginal ultrasound in women with endometrial cancer [16]. Therefore, the aim of our study was to determine the learning curve with the cumulative summation test for assessing myometrial infiltration in cases of endometrial cancer using transvaginal ultrasound (TVS) visualizing videoclips.

## 2. Materials and Methods

The training program was developed at one European University teaching hospital. The program design was presented to the University Educational Committee, who evaluated and approved the start of the program. Ethical review and approval were waived due to the retrospective and anonymous nature of the study.

The objective of this program was to train examiners with no or very little experience in ultrasound assessment of myometrial infiltration in women with proven endometrial cancer. Given that expert examiners have shown a diagnostic performance in terms of sensitivity of about 80% and specificity of about 84% when using examiner’s subjective impression [5], the objective was that trainees should achieve a diagnostic performance similar to those figures.

The training program consisted of three theoretical lectures (one hour each) addressing clinical aspects of endometrial cancer (epidemiology, oncogenesis, risk factors, clinical symptoms, diagnosis, staging and treatment), the rationale for assessing myometrial infiltration (from the gynecologic oncologist point of view) and how to perform ultrasound examination for assessing myometrial infiltration (ultrasound basis, anatomy, machine settings and methods for evaluating myometrial infiltration). This theoretical training was followed up by visualization of 10 videoclips explained by the trainer. The trainer was a gynecologist that had more than 25 years of experience performing gynecological ultrasound and had been involved in gynecologic oncology in a dedicated manner for 20 years. The method that was taught for assessing myometrial infiltration was based on subjective impression by watching the endometrial–myometrial border and assessing the symmetry of the myometrium in both longitudinal and transverse planes [7].

After this, each trainee had to visualize 45 videoclips of TVS assessment of the uterus from cases of endometrial cancer already treated surgically. The number of 45 videoclips was set arbitrarily. All real-time examinations and videoclip recordings had been performed by the trainer. Doppler assessment was not included in the videoclip. Videoclips were randomly selected by the trainer from the Department imaging database. The ratio of cases with <50% and ≥50% of myometrial infiltration was 4:1. The machine settings that were used were standard ones; they included the use of harmonics with a frequency of 3–7 MHz and medium gain. Videoclip duration was about 1–2 min. The time allowed to the trainee for visualizing each videoclip was up to five minutes. Definitive histology after uterus surgical removal was used as a reference standard. All trainees had to state whether myometrial infiltration was ≥50% or <50% (Figure 3 and Figure 4).

After visualizing each videoclip, the trainee was informed whether she/he was right or wrong regarding myometrial infiltration. In the case of mistakes, the videoclip was repeated with the trainer to learn why the mistake was made. This feedback was considered essential in the process of learning [17,18]. All trainees performed their evaluations blinded to each other. The videoclips were selected by the trainer in a ratio 4:1 for <50% and ≥50% infiltration cases according to the expected prevalence of deep myometrial infiltration in G1/G2 endometrioid carcinomas [19]. Cases were distributed randomly for trainee visualization. Trainees were blinded to this distribution. Training was performed on an individual basis because the trainees were blinded to each other during the process.

The learning curve cumulative summation (LC-CUSUM) test was used to assess each trainee’s learning curve [15]. Acceptable and unacceptable failure rates were set at 15% and 25%, respectively. These limits were chosen assuming that the pooled failure rate for an expert examiner could be around 15–25% taking into account both false-positive and false-negative results [5,20]. Type I (α) and type II (β) error rates were set at 0.1. CUSUM values were plotted on the *y*-axis, and the number of examinations was plotted on the *x*-axis. Horizontal lines were plotted at regular intervals on the *y*-axis defining *h*0 and *h*1 for the spacing between acceptable and unacceptable boundary lines, respectively. Competence is declared when the plot falls below two consecutive boundary lines [15].

## 3. Results

Five trainees (one staff radiologist and four fourth-year OB/GYN residents) participated in this study. All trainees had experience in performing TVS, with at least 200 examinations performed by each of them, but none of them had specific training on the assessment of myometrial infiltration in cases of endometrial cancer. All trainees completed the study blinded to each other in a sequential manner. The time spent by all trainees assessing the videoclips and debriefing with the trainer was three days (fifteen videoclips per day). We chose such a duration to avoiding trainee tiredness, which could have reduced attention during training.

LC-CUSUM graphics showed that four trainees reached competence at the 27th, 33rd, 35th and 36th examination, respectively (Figure 5, Figure 6, Figure 7 and Figure 8). Three of them kept the process under control after reaching competence, and one did not. The one who was not able to keep the process under control was the one who achieved competence at the 27th examination. We did not find a clear explanation for why this happened.

One trainee did not reach competence after visualizing the 45 videoclips (Figure 9).

## 4. Discussion

### 4.1. Summary of Findings

In this study, we have estimated the learning curve for assessing myometrial infiltration in patients with endometrial cancer using videoclips. We observed that competence for correctly interpreting the depth of myometrial infiltration in cases of endometrial cancer could be reached after visualizing 30–40 videoclips. We also observed a relatively low variability for reaching this competence. However, not all trainees reached competence or were able to maintain performance after apparently reaching competence.

### 4.2. Interpretation of Results

Training in ultrasound is essential to reach adequate skills [13,21]. It has been demonstrated that training improves clinical performance and patient-perceived quality of care [22,23]. Interpretation of images is also a relevant issue in the process of training [24,25]. Clearly, how long a given examiner has used transvaginal ultrasound in clinical practice is not the main factor that provides competence, but the number of cases evaluated for a given pathology or problem is a crucial one. Therefore, assessing the number of cases needed for achieving competence is an important issue, especially for developing training programs in gynecological ultrasound units.

In the case of transvaginal ultrasound evaluation of myometrial infiltration in endometrial cancer, there are very scanty data about the number of examinations needed to reach competence. Some reports indicated that a learning process exists, since diagnostic performance of the examiner increased with the number of cases evaluated. Alcazar and co-workers observed that sensitivity for detecting deep myometrial infiltration using the examiner’s subjective impression increased from 64% in the first 50 cases evaluated by the examiner to 92% when the same examiner had examined 50 to 100 cases [26]. Additionally, another study assessed inter-observer variability among expert and non-expert examiners for determining myometrial infiltration using videoclips [27]. This study observed that reproducibility was high among examiners with different levels of expertise, but diagnostic performance was higher in the case of expert examiners. Furthermore, a very recent multicenter study confirmed that examiners’ experience affected the diagnostic performance of TVS, but this was not case for MRI [28].

However, no study formally addressing the learning curve for ultrasound evaluation of myometrial infiltration in women with endometrial cancer has been reported until recently. We have shown that adequate skills for assessing myometrial infiltration in women with endometrial cancer may be acquired by interpreting two-dimensional real-time videoclips. Our data are in agreement with those reported by Xholli and colleagues [16]. These authors evaluated the learning curve for transvaginal ultrasound assessment of myometrial infiltration in women with endometrial cancer in a retrospective study. Two examiners participated in this study, and both were expert examiners. Although not specifically stated in their paper, apparently, the data used were derived from real-time ultrasound scanning and not videoclips. They used the Karlsson’s method for assessing myometrial infiltration and not the examiner’s subjective impression as we did. They established a failure rate at 25%, while our rate was 15%. According to their data, one examiner performed 42 examinations. This examiner reached competence after 29 examinations and kept performance under control afterwards. The other examiner performed 25 examinations and did not achieve competence. Comparing data from Xholli’s study and ours, it can be observed that in Xholli’s study, the examiner who achieved competence needed a slightly lower number of examinations than our trainees (29 versus 33–36). This can be explained by the method used for assessing myometrial infiltration. In Xholli’s study, the Karlsson’s method was used. This method utilizes the measurement of endometrial thickness and uterine anteroposterior diameter in the longitudinal plane. Both measurements have been demonstrated to have a short learning curve. On the contrary, we used the subjective examiner’s impression. This method might require more training, especially for examiners to become confident in her/his diagnosis.

On the other hand, Xholli and colleagues observed that one examiner did not achieve competence. We also observed that not all examiners may achieve competence in this field or may need more cases to achieve competence. This fact has also been observed in the ultrasound evaluation in women with pelvic endometriosis, as reported by Leonardi and colleagues [29]. This reinforces the concept that the same training program does not fit all trainees.

Finally, it is interesting to note that in Xholli’s study, the authors also assessed the learning curve to detect pelvic and para-aortic lymph nodes. They observed that the number of examinations required to achieve competence to detect positive lymph nodes in pelvic and para-aortic area was smaller than that for detecting deep myometrial invasion (13 examinations versus 29 examinations, respectively). This is rather surprising since examiners performing gynecological ultrasound are less accustomed to exploring pelvic and abdominal great vessels.

When developing a training program in ultrasound, a real-time training scenario is ideal [30,31]. However, this ideal scenario is not always available or feasible, or it might lead to long training periods. For this reason, simulations have been introduced in ultrasound training [22]. Three-dimensional ultrasound has also been advocated as a potential tool in ultrasound training, especially because it allows virtual navigation simulating real-time scanning. Additionally, it allows offline training, which can reduce the time needed for training [32,33]. However, three-dimensional ultrasound does not allow observation of the dynamic features of real-time ultrasound. For this reason, our group questioned whether the use of pre-recorded videoclips would be better than three-dimensional stored volumes for training. We tested this issue in a training program for ultrasound evaluation of adnexal masses [34]. We observed that when the trainee needs to assess dynamic features based on subjective impression, training was better when using two-dimensional pre-recorded real-time videoclips than using three-dimensional stored volumes. This was not the case when dynamic assessment is not needed, as in the case of uterine congenital anomalies [12]. For this reason, we selected videoclips for our training program related to the evaluation of myometrial infiltration in endometrial cancer. Somehow, in this particular topic, our data were in agreement with data reported by Green and colleagues related to the assessment of myometrial infiltration comparing 2D videoclips and stored 3D volumes [20]. In their study, these authors assessed inter-rater reliability and diagnostic accuracy of 15 expert examiners using either two-dimensional videoclips or three-dimensional volumes in a series of 58 patients with endometrial cancer. Although these authors did not assess the learning curve, they observed that inter-rater reliability was better using two-dimensional videoclips than three-dimensional volumes (kappa index for two-dimensional videoclips was 0.41 as compared to 0.31 for three-dimensional volumes). A similar result was observed for diagnostic accuracy (76% for two-dimensional videoclips as compared to 69% for three-dimensional volumes). These findings could support, indirectly, the idea that training with two-dimensional videoclips would be better than using stored three-dimensional volumes. Additionally, these authors observed that accuracy was correlated with the number of cases that each examiner annually assessed in their respective institutions, confirming that the number of cases assessed is important in training.

### 4.3. Strenthgs and Limitations

The main strength of our study is that this it is one of the first studies reported so far addressing learning curves for assessing myometrial infiltration in endometrial cancer using transvaginal ultrasound. Our data provide valuable information for those involved in ultrasound training and for those who want to be trained in this particular topic.

However, our study has limitations. The main limitation is that trainees did not perform real-time examinations but just interpreted videoclips. The learning curve of performing real-time examination by the trainee would probably need more cases. In fact, performance after training should be tested in a real-time stetting. Nevertheless, data from the study of Xholli and colleagues indicated that a similar number of cases were needed to achieve competence using apparently real-time examination (around 30 cases [16]), and, as stated above, the use of offline training has been demonstrated as a valid tool for training [21,22].

Another limitation might be the small number of videoclips for training because we also observed that one trainee did not reach competence. This might indicate that the training program needs to be improved, for example, by increasing the number of videoclips to be assessed and incorporating transvaginal ultrasound simulators that are available in the market [19]. This was also observed in the study by Xholli and colleagues [16].

Notwithstanding, we should bear in mind that in some cases, trainees trained for ultrasound examination are not able to acquire adequate skills. We observed this fact in one of our trainees. This was also observed in other studies addressing learning curve in ultrasound training in gynecology, such as in assessing pelvic endometriosis [29]. Additionally, our study was based on a small number of trainees, and incorporating more trainees could probably render different results regarding the average number of cases needed to reach competence. In fact, this is a limitation in most studies involving learning curve assessment in gynecologic ultrasound. In our opinion, this limitation precludes the possibility that our data are generalizable. However, in this regard, it is interesting to observe that many studies assessing gynecological training in different fields, such as evaluating pelvic endometriosis [34,35,36], ureteral evaluation [37,38], uterine sliding sign [39], ovarian endometrioma [40] and uterine adenomyosis [41], report similar data regarding the number of assessments needed to achieve competence (around 20 to 50 cases).

However, in spite of these limitations, we do think that our data might be relevant. Training in a real-life setting might be long. Considering that a center with a high volume of patients with endometrial cancer is that in which more 50 cases/year are treated [4], a training program in such a center would last at least several months. Using a training program based on image interpretation might reduce the time needed to reach competence in gynecological ultrasound, particularly for assessing myometrial infiltration in women with endometrial cancer, as shown in other disciplines [42].

## 5. Conclusions

In conclusion, our study suggests that visualization of videoclips could be useful for training sonographers in assessing myometrial infiltration by TVS in cases of endometrial cancer.

According to our data, about 30–40 cases would be needed to be trained. However, further studies with dedicated training programs are needed to confirm that e-learning can be an effective method for teaching and training in this field.

## Figures and Tables

**Figure 1 diagnostics-13-00425-f001:**
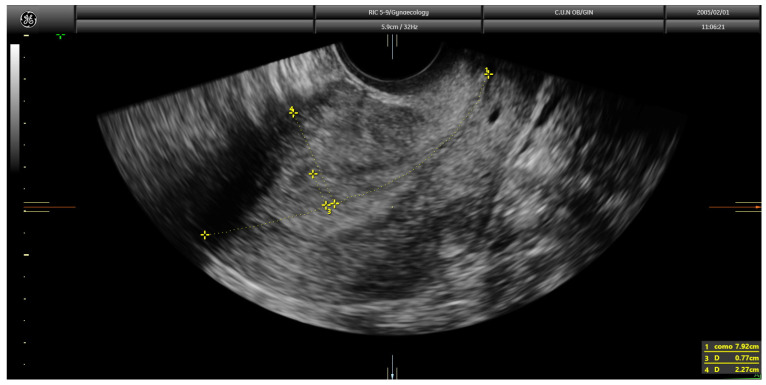
Transvaginal ultrasound showing a case of superficial infiltration in endometrial cancer using the Gordon’s method. The ratio of endometrial thickness/uterine anteroposterior diameter was 33.4%.

**Figure 2 diagnostics-13-00425-f002:**
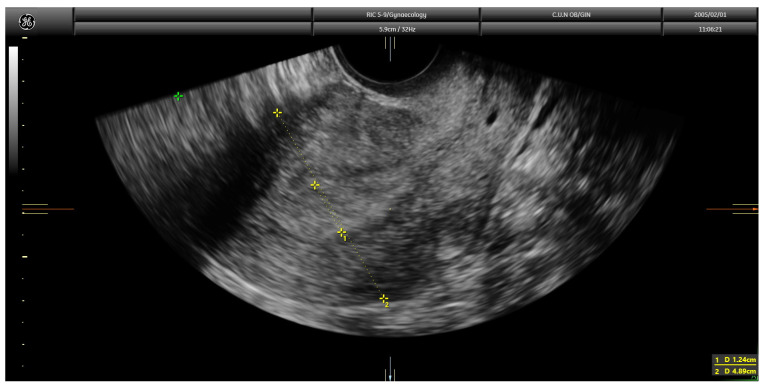
Transvaginal ultrasound showing a case of superficial infiltration in endometrial cancer using the Karlsson’s method. The ratio endometrial thickness/uterine anteroposterior diameter was 25.3%.

**Figure 3 diagnostics-13-00425-f003:**
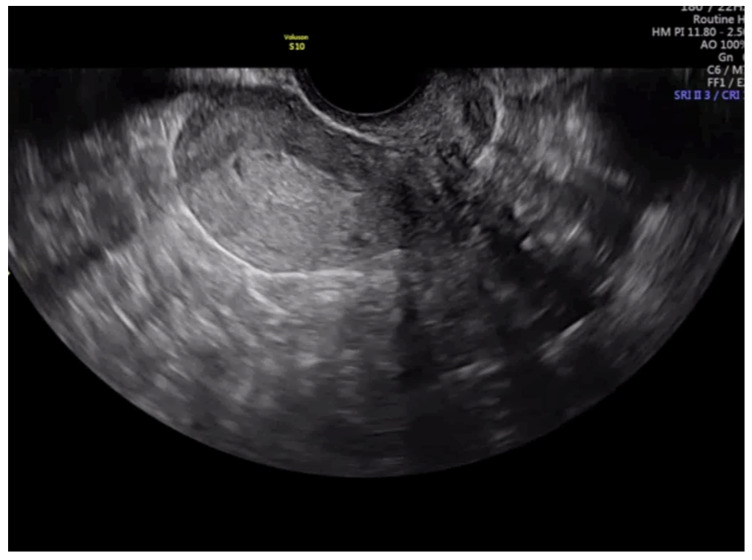
Transvaginal ultrasound showing a case of superficial infiltration in endometrial cancer according to subjective impression.

**Figure 4 diagnostics-13-00425-f004:**
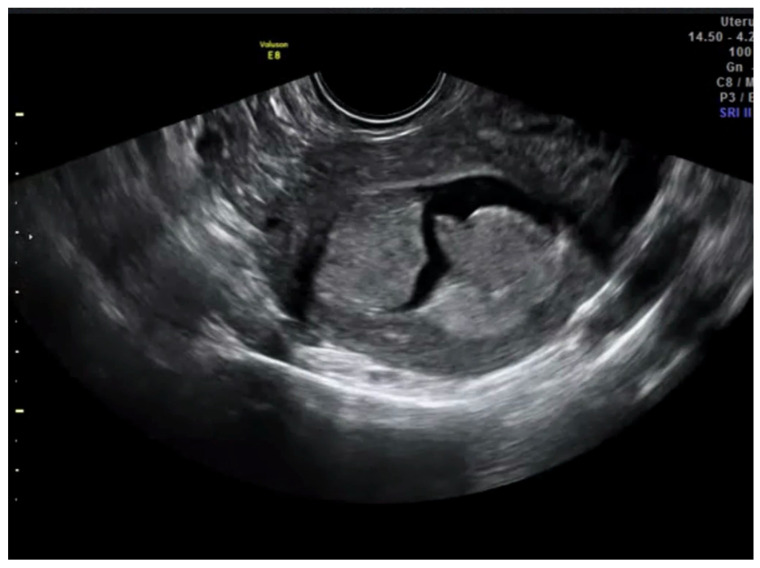
Transvaginal ultrasound showing a case of deep infiltration in endometrial cancer according to subjective impression.

**Figure 5 diagnostics-13-00425-f005:**
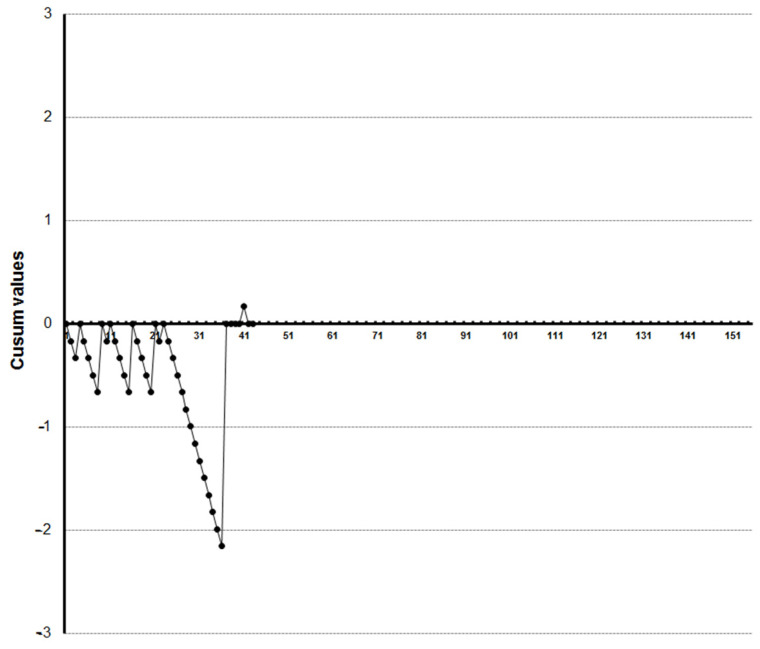
LC-CUSUM curve for trainee one.

**Figure 6 diagnostics-13-00425-f006:**
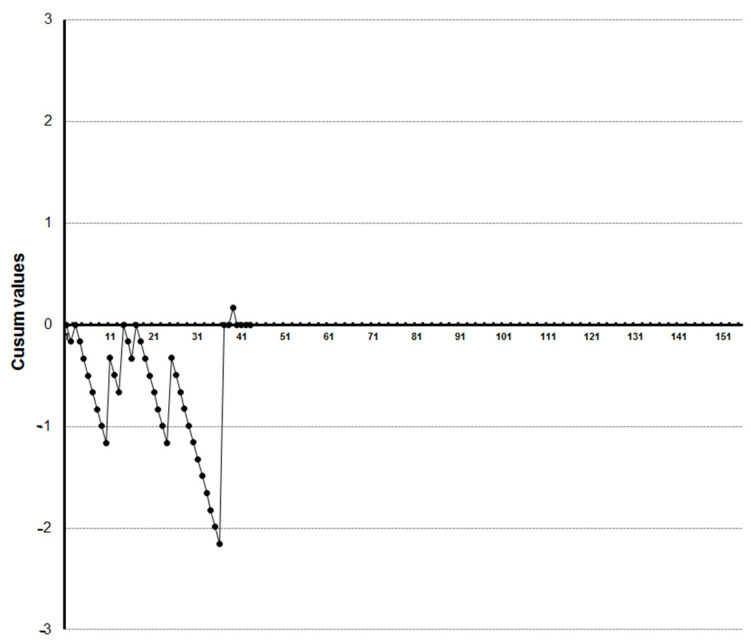
LC-CUSUM curve for trainee two.

**Figure 7 diagnostics-13-00425-f007:**
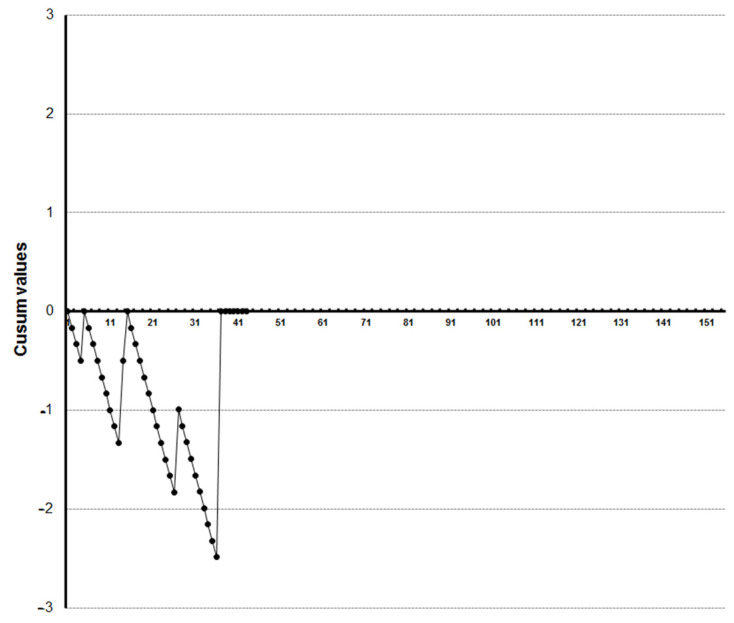
LC-CUSUM curve for trainee three.

**Figure 8 diagnostics-13-00425-f008:**
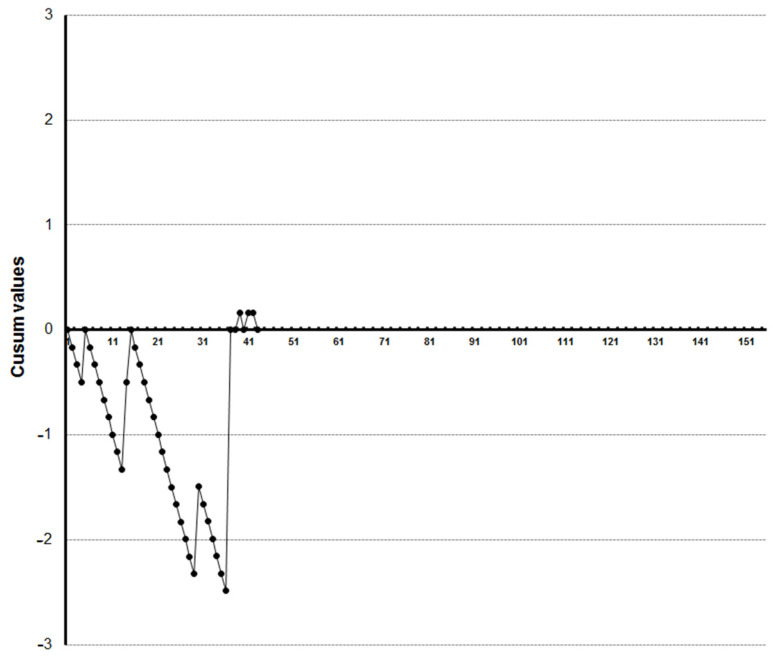
LC-CUSUM curve for trainee four. This trainee reached competence but did not keep it in the cumulative analysis.

**Figure 9 diagnostics-13-00425-f009:**
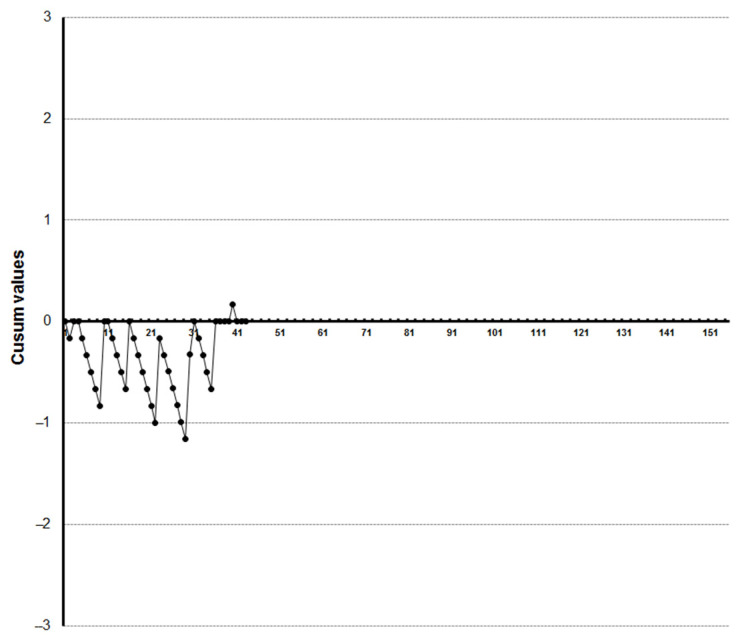
LC-CUSUM curve for trainee five. This trainee did not reach competence.

## Data Availability

Data are available upon reasonable request.

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
