# Peer review of "Learning Curve for Ultrasound Assessment of Myometrial Infiltration in Endometrial Cancer Visualizing Videoclips: Potential Implications for Training"

_diagnostics, 2023, doi:10.3390/diagnostics13030425_

Round 1

Reviewer 1 Report

Study is interesting because of training purpose in an important subspecialty -gynecological oncology. This training technique is indeed a good one, however the LC is actually different in real time examination when correct acquiring of images is mandatory.

Sample of trainees is very small therefore high quality conclusions are difficult to mention.

- Abstract section should be more concise and shortened to emphasize the mail idea.

- in introduction section, a picture demonstrating Karlsson’s and Gordon’s method (how to assess) would be useful - see Alcazar's manuscript fig 1 and 2 (ref 6)

- conclusion section should contain a take home message

Author Response

  1. Question: Study is interesting because of training purpose in an important subspecialty -gynecological oncology. This training technique is indeed a good one, however the LC is actually different in real time examination when correct acquiring of images is mandatory.
    1. Answer: Thanks for this comment. We agree with reviewer. For this reason we stress in the title this training is with video-clip visualization and also in the Discussion (See Strengths and Limitations section). No change made in the revised version.
  2. Question: Sample of trainees is very small therefore high quality conclusions are difficult to mention.
    1. Answer: We agree. This is already mentioned in the Discussion.
  3. Question: Abstract section should be more concise and shortened to emphasize the mail idea.
    1. Answer: Shortened from 320 words to 240 words.
  4. In introduction section, a picture demonstrating Karlsson’s and Gordon’s method (how to assess) would be useful - see Alcazar's manuscript fig 1 and 2 (ref 6)
    1. Answer: figures added
  5. conclusion section should contain a take home message
    1. Answer: We have modified conclusion section trying to mention take home messages

Reviewer 2 Report

Abstract: ”The objective of this study is to determine the learning curve (LC) for assessing myometrial infiltration in cases of endometrial cancer using transvaginal ultrasound (TVS)”. Why not for cervical invasion as well?

The Introduction is rather long and may be compressed in order not to lose the reader´s attention. Further, it would help the reader to illustrate Karlsson´s and Gordon´s methods with figures.

Introduction page 3, lines 100-103: The authors have only identified one study assessing the learning curve for evaluation of myometrial infiltration by TVS. However, a small study from 2015 (n=53) found no difference regarding MI between ultrasound experts and gynaecologists using ultrasound in their everyday work but having no experience in ultrasound staging of endometrial cancer (Eriksson LS, Lindqvist PG, Flöter Rådestad A, Dueholm M, Fischerova D, Franchi D, Jokubkiene L, Leone FP, Savelli L, Sladkevicius P, Testa AC, Van den Bosch T, Ameye L, Epstein E. Transvaginal ultrasound assessment of myometrial and cervical stromal invasion in women with endometrial cancer: interobserver reproducibility among ultrasound experts and gynecologists. Ultrasound Obstet Gynecol. 2015 Apr;45(4):476-82. doi: 10.1002/uog.14645. PMID: 25092412). In fact, in this study video clips were used. Since it indirectly copes with learning I suggest that you either mention the study here, or rather discuss it in Discussion.

In the PODEC multicentre study (n=259), TVS examinations were performed by 32 experienced gynaecologists (not ultrasound experts) after a short introduction/training (Palmér M, Åkesson Å, Marcickiewicz J, Blank E, Hogström L, Torle M, Mateoiu C, Dahm-Kähler P, Leonhardt H. Accuracy of transvaginal ultrasound versus MRI in the PreOperative Diagnostics of low-grade Endometrial Cancer (PODEC) study: a prospective multicentre study. Clin Radiol. 2023 Jan;78(1):70-79. doi: 10.1016/j.crad.2022.09.118. Epub 2022 Oct 19. PMID: 36270868). Specificity and sensitivity assessing deep myometrial invasion was 0.68. Although outperformed by MRI, the diagnostic accuracy of TVS was judged to be sufficiently adequate as a first-line technique. The learning curves of the TVS examiners were not studied specifically during the study period o three years. However, this was perfomed regarding an inexperinced MRI-reader (radiologist). The sensitivity increased from 0.59 to 0.75 and the specificity from 0.79 to 0.83. This is intersting, since MRI is generally regarded to be less operator depended compared to ultrasound.

M & M, page 3-5: More details are needed. How were the cases for the video clips selected? Who perfomed these examinations? How was the utrasound machine prestanda for these cases? Was color doppler and/or ultrasound contrast medium used?

Discussion: The studies mentioned (and cited) above could be discussed.

Discussion, page 8, lines 237-243: I suggest to delete this section. Assessing myometrial infiltration is an indirect method to estimate the risk of lymph node metastases and evaluating pelvic and para-aortic lymph nodes directly by ultrasound is indeed challenging (Eriksson LSE, Epstein E, Testa AC, Fischerova D, Valentin L, Sladkevicius P, Franchi D, Frühauf F, Fruscio R, Haak LA, Opolskiene G, Mascilini F, Alcazar JL, Van Holsbeke C, Chiappa V, Bourne T, Lindqvist PG, Van Calster B, Timmerman D, Verbakel JY, Van den Bosch T, Wynants L. Ultrasound-based risk model for preoperative prediction of lymph-node metastases in women with endometrial cancer: model-development study. Ultrasound Obstet Gynecol. 2020 Sep;56(3):443-452. doi: 10.1002/uog.21950. PMID: 31840873).

Discussion, page 9, lines 275-279, Strengths: I suggest that the fact that video clips were used in the learning process should be inserted into this sentence. It may be considered a strength that this ”simulation” technique seems to work well, not involving a patient in real time.

Thus, not performing real time examinations may not be considered a limitation (line 280-284).

The number of only 45 video clips may be considered a limitation. It is not perfectly clear that the skills were maintained once competence was considered reached for a trainee and we do not know if the individual who never reached competence would have done so if more video clips were used.

Author Response

  1. Question: Abstract: ”The objective of this study is to determine the learning curve (LC) for assessing myometrial infiltration in cases of endometrial cancer using transvaginal ultrasound (TVS)”. Why not for cervical invasion as well?
    1. Answer: Thanks for this comment. Interesting question. We agree that cervical invasion also needs training. However, to be honest we did not have video-clips of the cervix. So, we could not train this aspect. No change made.

  1. Question: The Introduction is rather long and may be compressed in order not to lose the reader´s attention. Further, it would help the reader to illustrate Karlsson´s and Gordon´s methods with figures.
    1. Answer: We agree. Figures added. However, we have decided not shortening the Introduction in order to achieve the minimum goal of 3000 words according to Editorial requirements.
  2. Question: Introduction page 3, lines 100-103: The authors have only identified one study assessing the learning curve for evaluation of myometrial infiltration by TVS. However, a small study from 2015 (n=53) found no difference regarding MI between ultrasound experts and gynaecologists using ultrasound in their everyday work but having no experience in ultrasound staging of endometrial cancer (Eriksson LS, Lindqvist PG, Flöter Rådestad A, Dueholm M, Fischerova D, Franchi D, Jokubkiene L, Leone FP, Savelli L, Sladkevicius P, Testa AC, Van den Bosch T, Ameye L, Epstein E. Transvaginal ultrasound assessment of myometrial and cervical stromal invasion in women with endometrial cancer: interobserver reproducibility among ultrasound experts and gynecologists. Ultrasound Obstet Gynecol. 2015 Apr;45(4):476-82. doi: 10.1002/uog.14645. PMID: 25092412). In fact, in this study video clips were used. Since it indirectly copes with learning I suggest that you either mention the study here, or rather discuss it in Discussion.
    1. Answer: Reviewer is right. We mention this paper in the Discussion. New reference added.
  3. Question: In the PODEC multicentre study (n=259), TVS examinations were performed by 32 experienced gynaecologists (not ultrasound experts) after a short introduction/training (Palmér M, Åkesson Å, Marcickiewicz J, Blank E, Hogström L, Torle M, Mateoiu C, Dahm-Kähler P, Leonhardt H. Accuracy of transvaginal ultrasound versus MRI in the PreOperative Diagnostics of low-grade Endometrial Cancer (PODEC) study: a prospective multicentre study. Clin Radiol. 2023 Jan;78(1):70-79. doi: 10.1016/j.crad.2022.09.118. Epub 2022 Oct 19. PMID: 36270868). Specificity and sensitivity assessing deep myometrial invasion was 0.68. Although outperformed by MRI, the diagnostic accuracy of TVS was judged to be sufficiently adequate as a first-line technique. The learning curves of the TVS examiners were not studied specifically during the study period o three years. However, this was perfomed regarding an inexperinced MRI-reader (radiologist). The sensitivity increased from 0.59 to 0.75 and the specificity from 0.79 to 0.83. This is intersting, since MRI is generally regarded to be less operator depended compared to ultrasound.
    1. Answer: We discuss this interesting paper in the Discussion. However, we have to note that sensitivity and specificity for MRI readers (expert and non-expert) in this study did not differ significantly.
  4. Question: M & M, page 3-5: More details are needed. How were the cases for the video clips selected? Who perfomed these examinations? How was the utrasound machine prestanda for these cases? Was color doppler and/or ultrasound contrast medium used?
    1. Answer: We have added this information
  5. Question: Discussion: The studies mentioned (and cited) above could be discussed.
    1. Answer: Done
  6. Question: Discussion, page 8, lines 237-243: I suggest to delete this section. Assessing myometrial infiltration is an indirect method to estimate the risk of lymph node metastases and evaluating pelvic and para-aortic lymph nodes directly by ultrasound is indeed challenging (Eriksson LSE, Epstein E, Testa AC, Fischerova D, Valentin L, Sladkevicius P, Franchi D, Frühauf F, Fruscio R, Haak LA, Opolskiene G, Mascilini F, Alcazar JL, Van Holsbeke C, Chiappa V, Bourne T, Lindqvist PG, Van Calster B, Timmerman D, Verbakel JY, Van den Bosch T, Wynants L. Ultrasound-based risk model for preoperative prediction of lymph-node metastases in women with endometrial cancer: model-development study. Ultrasound Obstet Gynecol. 2020 Sep;56(3):443-452. doi: 10.1002/uog.21950. PMID: 31840873).
    1. Answer:
  7. Discussion, page 9, lines 275-279, Strengths: I suggest that the fact that video clips were used in the learning process should be inserted into this sentence. It may be considered a strength that this ”simulation” technique seems to work well, not involving a patient in real time. Thus, not performing real time examinations may not be considered a limitation (line 280-284).
    1. Answer: We agree, we have modified Discussion.
  8. Question: The number of only 45 video clips may be considered a limitation. It is not perfectly clear that the skills were maintained once competence was considered reached for a trainee and we do not know if the individual who never reached competence would have done so if more video clips were used.
    1. Answer: We agree. Discussion added